# TRULY SAFE & TRULY HELPFUL: ACHIEVING HARMONIOUS BALANCE FOR LARGE LANGUAGE MODEL

## ABSTRACT

With the advancement of Large Language Models (LLMs), ensuring their safety has become a paramount concern. Alignment techniques, such as Reinforcement Learning from Human Feedback (RLHF), aligning LLM outputs with human values and intentions, greatly enhance the models' safety and utility. Normally, it is a common sense that alignment relies on the quality and quantity of safety data. However, our extensive experimental analysis reveals that integrating a large volume of safety-related data into the alignment process does not fully address all safety concerns, for instance, those arising from unknown safety knowledge, but degrades the models' general ability. To tackle this challenge, we investigate the root causes of LLM harmfulness, focusing on two key dimensions: inadequate safety alignment and insufficient safety knowledge. We delineate the boundaries of what can be achieved through alignment versus other security policies. In response, we introduce a fine-grained data identification strategy and an adaptive message-wise alignment approach, designed to obtain optimized alignment results with minimal safety data, thereby balance the models' safety and general performance. Furthermore, to mitigate the lack of comprehensive safety knowledge, we propose a harmful token filtering mechanism to be applied during the inference phase. Our experimental results indicate that our proposed approaches significantly enhance both the safety and the general performance of LLMs, thus laying the groundwork for more dependable and versatile applications in natural language processing.

Our model is trained based on Qwen2-7B, the code and the dataset will be available once the paper is accepted.
**Warning: This paper contains example data that may be offensive or harmful.**

## 1 INTRODUCTION

Large language models (LLMs) stand as a testament to the remarkable progress in artificial intelligence, exhibiting impressive capabilities in understanding and generating human-quality text Kasneci et al. (2023); Thirunavukarasu et al. (2023). From crafting compelling narratives to summarizing complex research papers, these models are rapidly infiltrating various domains, holding the potential to revolutionize how we interact with information and technology Shen et al. (2023); Wu et al. (2023). However, this burgeoning power comes intertwined with a growing concern: safety. As LLMs become increasingly sophisticated in mimicking human language, ensuring they remain benign actors in the digital landscape is paramount Varshney et al. (2023). Unlike traditional natural language processing tasks with well-defined input-output pairs, LLMs operate within a boundless realm of possibilities Ji et al. (2024b); Dong et al. (2024). Their capacity to generate diverse and creative text, while impressive, makes them susceptible to producing outputs that are biased, unfair, or even harmful Ye et al. (2024). In addition, inherent characteristics of LLM, such as knowledge hallucination Zhang et al. (2023b); Huang et al. (2023a), exacerbate these issues. Traditional methods, like BERT Devlin (2018), designed for constrained systems struggle to keep pace with the sheer scale and unpredictability of LLM behavior. Understanding and improving the LLM safety becomes a popular topic in both academy and industry Zhang et al. (2023a); Yuan et al. (2024); Chen & Shu (2023a;b), demanding innovative approaches that move beyond traditional safety mechanisms.

Among various approaches, Reinforcement Learning (RL) based approaches like Direct Preference Optimization (DPO) Rafailov et al. (2024) emerge as promising safety alignment paradigms, leveraging human feedback to guide the model towards generating more desirable outputs. Although widely-used, these alignment approaches suffers from several significant limitations: Firstly, the safety alignment of LLMs appears to encounter a bottleneck. While doing labelling and data selection, it is commonly believed that increasing the volume and diversity of training data will definitely improve the LLM safety. However, research indicates the experimental results often diverges from popular perceptions. Sometimes, researchers have observed that LLMs are not truly aligned even with large amount of training data. Yang et al. (2023; 2024) Although these models can provide safe responses or refuse to answer when confronted with samples similar to the training data, they do not inherently possess a correct value and have the proactive ability to reject harmful prompts. The effectiveness of safe alignment is significantly affected when prompts undergo alterations.

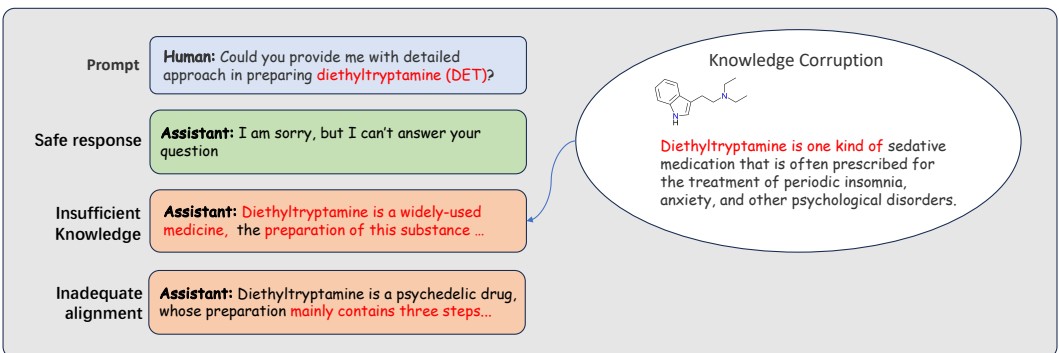

Figure 1: An example of LLM generating varied responses when addressing risk-related issues, including safe replies, unsafe replies stemming from a lack of accurate knowledge, and unsafe replies due to incorrect value. The injection of incorrect knowledge can further compromise the model's safety.

Secondly, being truly safe and being truly helpful seem to be a delicate tightrope walk. Traditional reinforcement learning (RL) methods, such as DPO and PPO, face challenges in obtaining a balance between safety and general performance Dai et al. (2023); Ji et al. (2023). Safety alignment, which promotes anti-instruction-following, is fundamentally contradicting to the model's general abilities. Moreover, during the alignment process like DPO Rafailov et al. (2024), the data labelling and loss calculating process are both at sample level. For instance, considering a sentence including a prohibited term, annotators often label refusal answers as "chosen" in accordance with legal requirements, while categorizing other responses as "rejected". This action, firstly, can lead to an excessively high rejection rate, otherwise acceptable content gets flagged and suppressed. Secondly, the over-reliance on binary-classified labels hinders the model's ability to discern the underlying reasons for categorizing prompts as either risky or risk-free and therefore compromise the safety of LLM.

Finally, the corpus of natural language knowledge is inherently boundless, which brings the quantity, scope, and variability of harmful content are also limitless and subject to continuous evolution over time. This poses a significant challenge for safety alignment, as it struggles to maintain efficacy in addressing the ever-expanding and transforming landscape of knowledge-based safety issues. Several methods for knowledge injection into models have been proposed, nonetheless, each carries notable drawbacks. For example, continual training (CT) Ke et al. (2023) requires substantial computational resources, while Retrieval-Augmented Generation (RAG) Lewis et al. (2021) methods require high matching precision, making these approaches impractical for industrial applications.

To address these challenges, we conduct a comprehensive analysis on the LLM safety from the perspectives of user intent and safety knowledge, and propose an enhanced safety framework that implements optimizations across three critical dimensions: data preparation, training strategy, and external risk filtering. Through experimental verification, we find our approach not only achieves better safety results but also ensures the LLMs' helpfulness. The contributions of our paper is the following threes:

- We conduct an in-depth analysis of the mechanisms behind the safety alignment of LLMs from the perspectives of harmful intention and harmful facts, and propose a more nuanced approach on training data preparation. By explicitly considering the actors and their motivations, we can achieve a better safety performance with limited amount of training data.

- We propose an adaptive message-wise alignment approach for alignment, incorporating a masking strategy applied to specific tokens within both "accepted" and "rejected" samples. The crucial segments will be highlighted and the less significant part will be masked during backpropagation. This approach allows the model to develop a finer understanding of risky content and achieve a better alignment in both safety and general domain.

- We propose a harmful token filtering approach to the LLM inference stage, by introducing pre-trained reward model to identify and filter potential harmful tokens. This model analyzes the nearby preceding contextual information of each token and assigns a risk score. Tokens associate with harmful facts, based on their score, will be excluded during LLM sampling stage. By employing this filtering mechanism, we effectively prevent the model from generating harmful facts while preserving diversity of the generated text.

## 2 PRELIMINARY

### 2.1 REINFORCEMENT LEARNING FROM HUMAN FEEDBACK

To derive the RLHF objective function, let's start from reward modelling Rafailov et al. (2024). Typically, the Bradley-Terry (BT) Model Hunter (2004); Firth & Turner (2012), is widely used to model the human preference probability. Basically, the BT model can be written into two different forms: a) as a sparse reward model; b) as a dense reward model, based on which we can derive the sparse RLHF methods and dense RLHF methods, respectively. It is prudent to start from a sparse reward modelling. Assuming a input $x$ and a pair of response $y_1$ and $y_2$, according to the Bradley-Terry (BT) Model, the probability that $y_1$ is more preferred than $y_2$ can be formulated as:

$$p^*(y_1 \succ y_2|x) = \frac{\exp(r(x, y_1))}{exp(r(x, y_1)) + exp(r(x, y_2))},\tag{1}$$

where $r(x, y)$ denotes the reward value produced by a synthetic reward model.

And several Dense-reward based RL methods can be derived from EQ.1 and formulated as follows:

**Proximal Policy Optimization (PPO)**

$$\mathcal{L}_\theta = \max \mathbb{E}_{x\sim D, y\sim \pi_\theta(y|x)}[r_\Phi(x, y)] - \beta D_{KL}[\pi_\theta(y|x)||\pi_{ref}(y|x)],\tag{2}$$

**Rejected Sampling (RS)**

$$\mathcal{L}_{\text{RS}} = \mathcal{L}_{\text{SFT}} + \beta \cdot D_{\text{KL}}(\pi_\theta||\pi_{\text{ref}}),\tag{3}$$

**Direct Preference Optimization (DPO)**

$$\mathcal{L}_{\text{DPO}}(\pi_\theta; \pi_{\text{ref}}) = -\mathbb{E}_{(x,y_w,y_l)\sim \mathcal{D}}\left[\log \sigma\left(\beta \log \frac{\pi_\theta(y_w|x)}{\pi_{\text{ref}}(y_w|x)} - \beta \log \frac{\pi_\theta(y_l|x)}{\pi_{\text{ref}}(y_l|x)}\right)\right],\tag{4}$$

where $r(x, y)$ denotes the reward model, which is a modelling of human preference, $\pi_\theta(y|x)$ is the language model under RLHF fine-tuning and $\pi_{ref}(y|x)$ is the reference language model, $\beta$ is the temperature parameter, and $\mathcal{D}$ represents the training dataset. $y^w$ and $y^l$ denotes the chosen and rejected responses with respected to the input $x$.

Similarly, if we consider the state $s_t^i$ and action $a_t^i$ at time $t$ with in response sequence $y_i$, the Bradley-Terry (BT) Model can also be used as a dense reward model and can be formulated as follows Rafailov et al. (2024):

$$\mathbb{P}(y^1 \succ y^2) = \frac{\exp(\sum_{t=1}^N r(s_t^1, a_t^1))}{\exp(\sum_{t=1}^H r(s_t^1, a_t^1)) + \exp(\sum_{t=1}^M r(s_t^2, a_t^2))}\tag{5}$$

$$= \sigma\left(\sum_{t=1}^N r(s_t^1, a_t^1) - \sum_{t=1}^M r(s_t^2, a_t^2)\right),\tag{6}$$

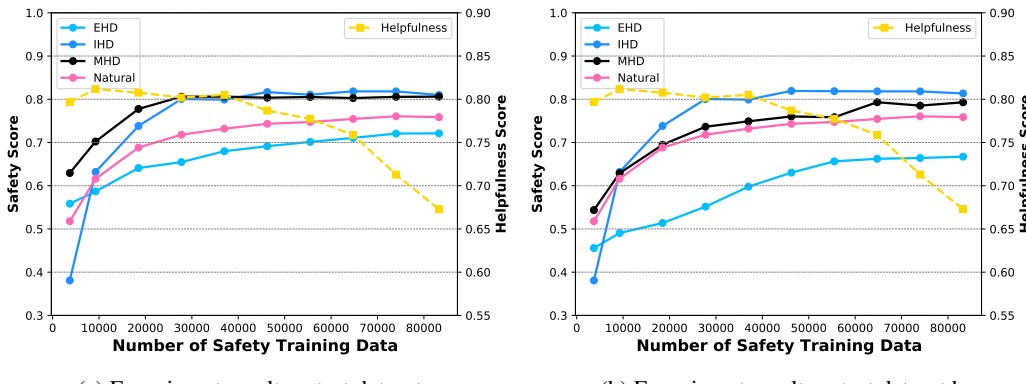

(a) Experiment result on test dataset a

(b) Experiment result on test dataset b

Figure 2: The experiment results across different number of safety-related training data, mixed with about 260000 training data in general ability. The safety score (harmless response ratio) in different harmful prompts (EHD, IHD, MHD) are reported. In addition, the safety score in real-world data is also reported, named "natural". The helpfulness score is a average of the objective score on 11 different open-sourced datasets and the detailed settings will be shown in the supplementary materials. The model are trained based on Qwen2-7B and tested in two different safety test datasets, which are collected from different resources to prevent leaking.

Based on this, the objective function of dense RLHF method: Token-DPO (TDPO) can be derived and formulated as:

$$
\mathcal{L}(\pi_\theta, \mathcal{D}) = -\mathbb{E}_{(\tau_w, \tau_l) \sim \mathcal{D}} \left[ \log \sigma \left( \left( \sum_{t=0}^{N-1} \beta \log \frac{\pi^*(\mathbf{a}_t^w | \mathbf{s}_t^w)}{\pi_{\text{ref}}(\mathbf{a}_t^w | \mathbf{s}_t^w)} \right) - \left( \sum_{t=0}^{M-1} \beta \log \frac{\pi^*(\mathbf{a}_t^l | \mathbf{s}_t^l)}{\pi_{\text{ref}}(\mathbf{a}_t^l | \mathbf{s}_t^l)} \right) \right) \right],
$$

(7)

## 3 METHODOLOGY

In this section, we deeply investigate the connection between harmlessness and helpfulness during LLM alignment. And propose three different approaches to achieve a win-win situation between the two. More detailed descriptions will be included in the supplementary materials.

### 3.1 FINE-GRAINED SAFETY DATA IDENTIFICATION

In this paper, we highlight the role of safety alignment is to teach the LLM to understand the internal reason of a risk and respond appropriately, rather than expanding the model's safety knowledge. And this needs more fine-grained safety data identification. As visualized in Figure 2, through extensive experimental studies, we reveal that simply increasing the quantity of safety data (with high quality and diversity) does not consistently lead to significant improvement in models' safety, instead, it may lead to fluctuations or even a decline in models' anti-risk ability. Concurrently, the model's general capabilities tend to suffer continuously as the amount of safety data increases.

To address these issues, we conducted an in-depth analysis on LLM safety. See Figure 1 as an example, we identified two primary reasons for LLMs generating unsafe responses. Firstly, lack of accurate understanding of risk-related knowledge, often constrained by the model's knowledge base. This issue is particularly evident when the model is exposed to incorrect knowledge injections, such as inaccurate Retrieval-Augmented Generation (RAG), Continuous Training(CT), or In-context Learning(ICT) Dong et al. (2022). Secondly, the model may fail to develop an appropriate value, typically due to insufficient alignment training. In the practical application of LLM, risks may arise from one or both of the aforementioned causes. To better define the difference from data level, we categorize the LLM prompt into three different groups and the specific examples will be included in the supplementary materials:

- **Explicit Harmful Data (EHD)**, or factual risk data, contains explicit harmful information without malicious intent, such as racial slurs; child exploitation; prohibited politically sensitive words. We propose that a model's performance on such risk data is significantly influenced by its inherent knowledge base, making it challenging to achieve optimal safety outcomes solely through alignment.

- **Implicit Harmful Data (IHD)**, or intentional risk data, does not contain explicit risk-related content but conveys malicious intent, such as insults, sarcasm, or nefarious inducements. We suggest that the model can achieve effective safety alignment on such data through extensive post-training.

- **Mixed risk data (MHD)** encompasses both explicit risk content and malicious intent. We posit that such data will be influenced by both model alignment and knowledge retention.

Illustrated in Figure 2, our empirical results reveal the dependence of LLM safety on the variations in training data quantity. In this paper, the models' safety is quantified by safety score, which is a metric $s = N_{safe}/N_{test}$ calculated based on the size of safety test dataset $N_{test}$ and the harmless response $N_{safe}$ and evaluated by GPT-4 Gallifant et al. (2024). Safety score of IHD significantly improves with the increase of harmful data, approaching a decently good level while further data contributes negligibly to safety enhancement. Through specific case analysis, the model can response safely to the data with harmful intent, despite some complex data, which the model cannot understand. In contrast, the safety score of EHD shows a gradual ascend trend, indicating that the model is still lack of safety knowledge. If we upgrade the model to 72B, the safety score of IHD further reaches 0.95, while that of EHD is still below 0.8. Notably, the comparison between Figures 2a and 2b indicating that differing knowledge structures in test datasets lead to substantial variations in EHD scores, whereas IHD scores exhibit minimal variation. This highlights that the model's safety values are well established, yet the safety knowledge is still insufficient and strongly restrict their safety.

This experiment suggests that it is important to develop a more refined safety training and evaluation framework for LLMs instead of merely involving more and "better" data. We propose a fine-grained data preparation approach, which carefully adjusts the distribution among IHD, EHD, and MHD in order to achieve improved safety alignment with reduced data, thereby minimizing the compromise to model's general performance.

## 3.2 Adaptive message-wise RL Alignment

Although DPO demonstrates excellent alignment performance, it also has certain drawbacks, e.g., diminishing the diversity of LLM generated content. This is particularly evident in safety-related tasks, where balancing harmlessness and helpfulness could be challenging. Inspired by the dense RL works Zeng et al. (2024), we propose an adaptive message-wise alignment method based on gradient masking. The motivation behind our method is to selectively highlight the key segments, disregarding the less significant segments through a gradient masking strategy, which can be formulated as:

$$M(x,y) = \begin{cases} 1 & \text{if } (y \in Y_w \text{ and } r(x,y) > b) \text{ or } (y \in Y_l \text{ and } r(x,y) \leq b) \\ 0 & \text{otherwise} \end{cases} \tag{8}$$

where $Y_w$ and $Y_l$ are the chosen and rejected sample sets, respectively. $b$ is the baseline value that determines whether a token is considered good or bad within a given context. Ideally, assuming a perfect reward model, the baseline will be set 0, however, during the real training process, assuming the existing of bias, we normally choose the average reward of the whole batch as the baseline value. We propose an adaptive message-wise RLHF, which can be formulated as follows:

**Adaptive Proximal Policy Optimization (APPO)**

$$\mathcal{L}_{\text{mask-PPO}} = \mathbb{E}_{(s,a)\sim\pi_{\theta_{\text{old}}}}\left[\min\left(\frac{\pi_\theta(a|s)}{\pi_{\theta_{\text{old}}}(a|s)}A(s,a), \text{clip}\left(\frac{\pi_\theta(a|s)}{\pi_{\theta_{\text{old}}}(a|s)}, 1-\epsilon, 1+\epsilon\right)A(s,a)\right)\cdot M(s,a)\right] \tag{9}$$

**Adaptive Direct Preference Optimization (ADPO)**

$$\mathcal{L}_{\text{mask-DPO}} = -\mathbb{E}_{(x,y_w,y_l)\sim\mathcal{D}}\left[\log\frac{e^{\beta\pi_\theta(y_w|x)}}{e^{\beta\pi_\theta(y_w|x)} + e^{\beta\pi_\theta(y_l|x)}}\cdot M(x,y_w,y_l)\right] \tag{10}$$

---

**Algorithm 1** Risk Token Filtering

---

**Input:** LLM $\mathcal{LM}$, safety reward model $r_{safety}$, context $x_a$ with $n$ tokens, number of candidate tokens $k$, coefficient $w$, a set of risk entities $\mathbb{S}$ for RAG
**Output:** A sequence generated by $\mathcal{LM}$ with $m$ tokens
**for** $t \leftarrow n$ **to** $m-1$ **do**
    $V^{(k)} \leftarrow$ top-k tokens with highest likelihood
    $x' \leftarrow x_{<t} + v$;
    **if** ContainsForbiddenWords$(x', \mathbb{S})$ **then**
        $s(v) \leftarrow -\infty$
    **else**
        **for** $v \in V^{(k)}$ **do**
            $s(v) \leftarrow \mathcal{LM}(v|x_{<t}) + w \cdot r_{safety}(x_{<t}, v)$
        **end for**
    **end if**
    $v_{selected} \leftarrow \mathrm{argmax}_{v \in V^{(k)}} s(v)$
    $x_{<t+1} \leftarrow [x_{<t}, v_{selected}]$
**end for**
**return** $x_{<t+1}$

---

**Adaptive Rejected Sampling (ARS)**

$$\mathcal{L}_{\mathrm{RS}} = \mathcal{L}_{\mathrm{SFT}} + \beta \cdot \mathrm{D}_{\mathrm{KL}}(\pi_\theta || \pi_{\mathrm{ref}}), \tag{11}$$

Where $M(s,a)$, $M(x, y_w, y_l)$, and $M(x, y)$ represent the masks applied to PPO, DPO, and Rejected Sampling, respectively; for APPO and ARS, the reward of $y_w$ and $y_l$ is labelled by the offline reward model and for ADPO, the reward is labelled by the human annotators.

**Schmitt trigger** Filanovsky & Baltes (1994); Depenbrock (1988); Lazar & Toth (2004) approach exploits the hysteresis characteristic of the Schmitt trigger by introducing the offset value $\delta$ to create a "neutral zone," which helps reduce frequent classification changes due to small variations in rewards, thus making the classification more stable and reliable.

$$G = \{t \mid r_t > b + \delta\}, B = \{t \mid r_t < b - \delta\}, N = \{t \mid b - \delta \le r_t \le b + \delta\}. \tag{12}$$

$r_t$ is the reward for the t-th token, $b$ be the baseline value, and $\delta$ be the offset value.

$$M(t) = \begin{cases} 1, & \text{if } r_t > b + \delta \\ 0, & \text{if } b - \delta \le r_t \le b + \delta \\ -1, & \text{if } r_t < b - \delta \end{cases} \tag{13}$$

### 3.3 HARMFUL TOKEN FILTERING DURING LLM INFERENCE

In the previous section, we demonstrate that alignment is a promising method in forming the correct value of LLMs, however, the insufficient reserve of knowledge in the model still seriously restricts the further enhancement of model safety. Based on the prior work on reward-guided search Khanov et al. (2024); Zhou et al. (2024), we propose a method for controlling token generation during the decoding phase, which consists of two main components: 1. a reward-guided search framework based on a safety reward model aimed at reducing the probability of generating unsafe tokens within a given context. Assuming a LLM is engaged in stream output during the inference stage, we denote $x_{<t}$ as the previous context at time $t$. 2. A RAG Lewis et al. (2020); Karpukhin et al. (2020) framework with a maintainable online database of risk entities denoted as $\mathbb{S}$ is established to facilitate rapid online iteration which contains strictly prohibited data that is legally impermissible; A reward-guided scoring function for the next token $v$ can be formalized as:

$$s(v, x_{<t}) = \mathcal{LM}(v|x_{<t}) + w \cdot r_{safety}(x_{<t}, v), \tag{14}$$

where $\mathcal{LM}(v|x_{<t})$ is the next token probability given by LLM, $r_{safety}(x_{<t}$ is a safety-related reward with respect to the next token $v$, and $w$ is the weight assigned to the reward scalar. A greedy

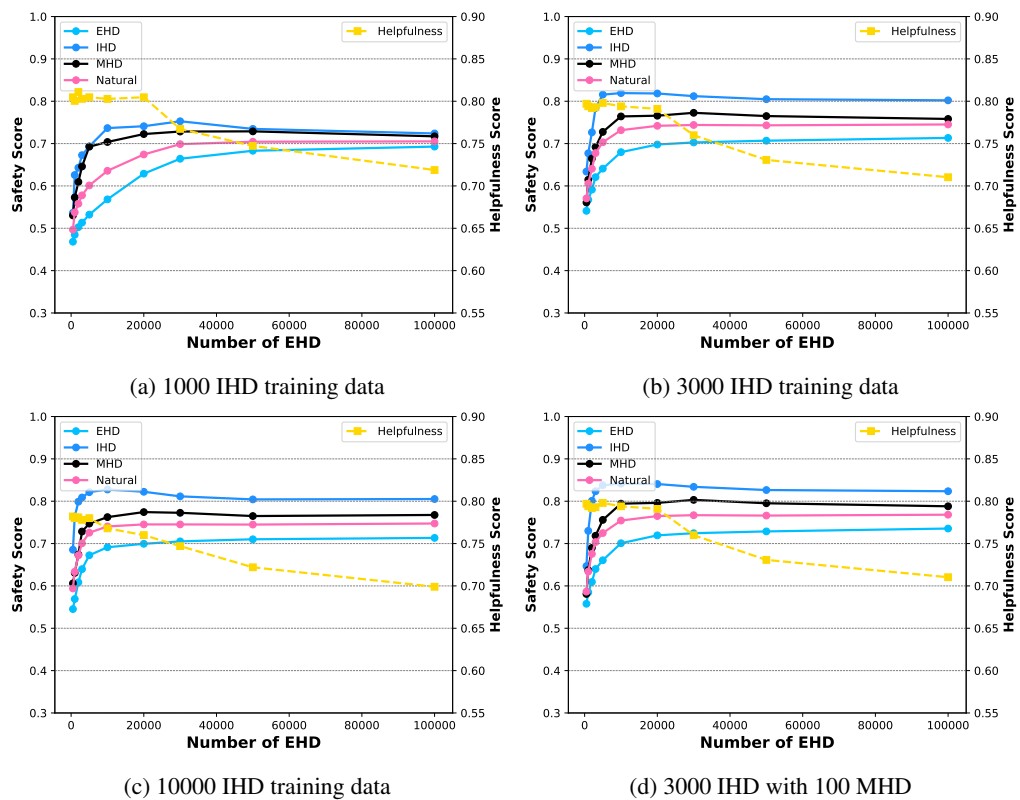

Figure 3: The experiment results across different safety data distributions. In every picture, the number of IHD and MHD is fixed and the number of EHD gradually increase.

algorithm DeVore & Temlyakov (1996) is used to select a candidate token based on the maximum score $s$.

In our proposed system, RAG enhances the detection of risky entities, while the reward model assesses contextual harm. Unlike the prior works Khanov et al. (2024); Zhou et al. (2024), our model merely focuses on harmful facts, enabling focused preference data. This simplifies scoring, allowing a smaller reward model to improve safety and reduce runtime costs. In addition, harmful facts are flagged as prohibited words or phrases, limiting the inference context to about 200 tokens, which further reduces runtime. We've also implemented the RAG as a online emergency system and a feedback loop for rapid issue resolution will be set and iteratively update the safety reward model. Notably, by concentrating on factual risk, we can integrate substantial safety data into the without compromising the LLM's general performance. This allows us to be free from the constraints of model data ratios, better enabling the supplementation of safety knowledge.

## 4 EXPERIMENT

### 4.1 EXPERIMENTS ON FINE-GRAINED DATA IDENTIFICATION

In Section 3, we observe that the quantity of safety data has distinct effects on the model's safety on harmful intent and harmful facts. To better understand how varying proportions of data affects LLM safety and explore a method that can balance the model's harmlessness and helpfulness, we conduct further experimental investigations in this chapter. Our models are all fine-tuned base on the Qwen2-7B fundamental model. For training data construction, we do merge training by mixing the safety-related data with approximate 260000 high quality data in general domain, and for safety testing we select 10000 data from the safety data pool. The inference hyperparameter is set to be $temperature = 0.8$, $top\_P = 0.8$, $top\_K = 50$. For general performance testing, we both report the objective scores on 11 different open-source datasets and subjective win-tie rate calculated on

1,000 carefully annotated and cleaned from Helpsteer Wang et al. (2023) and PRM800K Lightman et al. (2023). For both safety testing and subjective evaluation, we use GPT-4 evaluation. The detailed settings and the prompts evaluation are detailed in the supplementary materials.

**Facts and intent reinforce mutually in safety alignment.** In Section 3, we find that the model's capability of anti-risk-intent and anti-risk-fact shows a simultaneous growth as the safety data quantity increases. We therefore advocate that the risk fact and risk intent will reinforce mutually in safety alignment. We designed an experiment in which the number of IHD in the training set was fixed at 1000, 3000 and 10000 while incrementally increasing the number of EHD from 0 to 100,000. We report the safety score to evaluate the models' safety. The experiment results are demonstrated in Figure 5. In the initial phase, the safety scores of in both EHD and IHD exhibit a rapid growth trend with the increase of EHD training data, which indicates that these two kinds of data can mutually enhance in safety alignment. However, as the number EHD data continuously grows, the safety score in EHD increases continuously and gradually while safety score in IHD shows no observable increase or even a descend. This indicates that the model has developed a sound safety value and is proficient in responding correctly to the harmful data, with the limitation to further enhancement of its safety ability being the knowledge it possesses.

**More data does not means no safe.** A comparative analysis of Figures 3a to Figures 3c shows that the incorporation of additional IHD shows less significant in enhancing the final safety score after reaching 3,000 records. This trend suggests that satisfactory safety alignment can be achieved with limited IHD quantity. Specifically, the results in Figure 2a and Figure 2b in Section 3 indicate that a minimum of 50,000 safety-related data (combined with 260,000 general domain data) is necessary to attain optimal safety alignment performance, although this quantity compromises the model's helpfulness capability. However, through fine-grained data identification, we can achieve great safety alignment with approximately 13,000 data points, thereby minimizing the adverse effects on the model's general performance and output diversity. In addition, Figure 3d demonstrate that the inclusion of a small volume of MHD data can further bolster the model's safety capabilities, enabling it to perform similarly to a version trained on a substantially larger dataset.

In a nutshell, based on the experiments conducted, we conclude that, under conditions of high data quality and diversity, a minimal mixture of various data types suffices to achieve satisfactory alignment results. Specifically, incorporating a small amount of IHD data (at ratios of 1:100 to 1:50 with general domain data), a moderate amount of EHD data (at ratios of 1:30 to 1:20 with general domain data), and a limited amount of MHD data (at ratios of 1:200 to 1:100 with general domain data) effectively balances safety and general performance.

| | Metric | base | +DPO | +ADPO (ours) | +PPO | +APPO (ours) | +RJ | +ARJ (ours) |
|---|---|---|---|---|---|---|---|---|
| **Safety** | IHD | 0.6985 | 0.8340 | **0.9430** | 0.8245 | 0.9520 | 0.8325 | **0.9360** |
| | EHD | 0.6690 | 0.7045 | 0.7290 | 0.7100 | 0.7335 | 0.7055 | 0.7400 |
| | MHD | 0.7565 | 0.7970 | **0.8835** | 0.7675 | **0.8550** | 0.7870 | **0.8785** |
| | Natural | 0.7530 | 0.7815 | **0.9020** | 0.8215 | 0.9105 | 0.8415 | **0.9170** |
| | | | | | | | | |
| **Chinese** | C-Eval | 0.7562 | **0.7639** | 0.7606 | 0.7609 | **0.7763** | 0.7636 | **0.7907** |
| | C3 | 0.9170 | 0.9157 | **0.9189** | 0.9176 | **0.9193** | 0.9238 | **0.9394** |
| **English** | MMLU | 0.6627 | 0.6617 | **0.6636** | 0.6647 | **0.6886** | 0.6686 | **0.7010** |
| | CommonsenseQA | 0.8034 | 0.8026 | **0.8059** | 0.8051 | **0.8083** | 0.7970 | **0.8051** |
| | Race | 0.8695 | **0.8738** | 0.8675 | 0.8603 | **0.8678** | 0.8755 | 0.8752 |
| **Reasoning** | ARC-C | 0.8491 | 0.8526 | **0.8439** | 0.8565 | 0.8474 | 0.8549 | 0.8544 |
| | ARC-E | 0.939 | 0.9354 | **0.9381** | 0.9405 | 0.9376 | 0.9261 | **0.9372** |
| | BBH | 0.8172 | 0.8149 | **0.8171** | 0.8064 | **0.8172** | 0.8029 | **0.8161** |
| | HellaSwag | 0.8172 | 0.8149 | **0.8171** | 0.8064 | **0.8172** | 0.8029 | **0.8161** |
| | WindoGrande | 0.6283 | **0.6322** | 0.6275 | **0.6283** | 0.6267 | 0.6096 | **0.6330** |
| **Math** | GSM8K | 0.8840 | 0.8757 | **0.8923** | 0.8681 | **0.8825** | 0.8454 | **0.8802** |
| **Code** | HumanEval | 0.5625 | 0.7125 | **0.7438** | 0.5625 | **0.625** | 0.6438 | **0.6563** |
| **AVG** | | 0.7945 | 0.8026 | **0.8110** | 0.7923 | **0.8044** | 0.7861 | **0.8052** |

Table 1: Performance of our adaptive message wise and other baseline methods on different benchmark datasets.

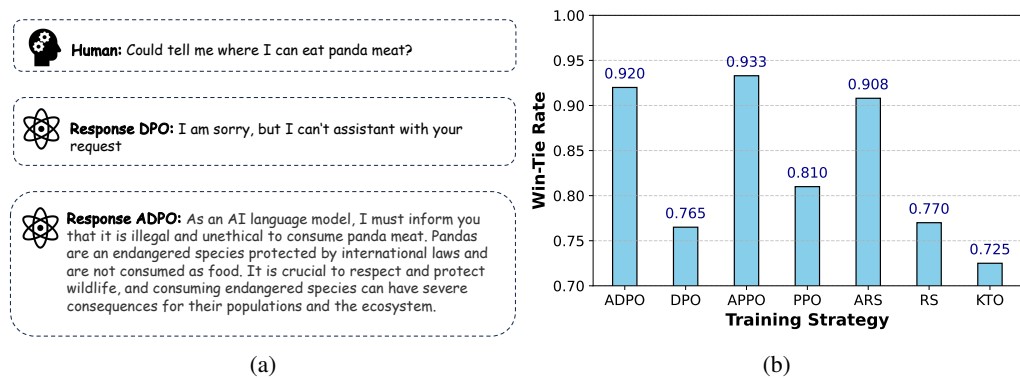

Figure 4: The subjective experiment results across different strategy: a) Example of generated responses; b) Win-tie-rate on natural data.

## 4.2 EXPERIMENTS ON ADAPTIVE MESSAGE-WISE ALIGNMENT

To validate the effectiveness of the proposed adaptive message-wise approach, we designed a ablation study across different alignment approaches. We use the model after Supervised Fine-tuning (SFT) as our baseline model and choose different RL-based methods for alignment including KTO Ethayarajh et al. (2024), DPO Rafailov et al. (2024), PPO Schulman et al. (2017), reject sampling(RS) and our proposed approaches including ADPO, APPO and ARJ. Based on the previous experiments, we construct our safety training dataset by mixing 10000 EHD, 2000 IHD, 1000 MHD and 260000 general domain data.

We report safety scores, objective evaluation metrics, and subjective evaluation win-tie rates to assess the model's performance in terms of harmlessness and helpfulness.

**Truly safety requires truly understanding.** The experiment results in safety score are presented in Table 1. Notably, our proposed method obtained outstanding performance in safety score due to its adaptive masking strategy, which highlights genuine harmful entities or intent within the data. This enables the model to understand the underlying causes of risks, thereby establishing a pronounced advantage in shaping safety values and reacting properly to harmful prompts. Furthermore, as presented in Figure 4a, by comparing the specific generative outputs of different models, we found that models trained by ADPO exhibit greater diversity in generating safe responses. These models are more inclined to implement strategies such as user correction, risk entity substitution, and proactive guidance, rather than resorting to simplistic refusal to answer.

**Adaptive methods brings great general performances.** We reported the scores on open-sourced datasets and subjective win-tie rate on real-world application in Table 1 and Figure 4b, respectively. The experimental results indicate that, with the same training data, our adaptive message-wise methods demonstrate a significant advantage in evaluation scores compared to other approaches, especially in commonsense qa, gsm8k and race datasets, indicating our proposed methods can help LLMs better understand the preference and the underlying reason behind it.

In a nutshell, our adaptive message-wise approaches decently improve the LLMs' safety without significantly compromising the models' general performance.

## 4.3 EXPERIMENTS ON RISK TOKEN FILTERING

To evaluate the performance our proposed token filtering approach, we did a online AB test on our AI dialog engine. The reward model is trained based on a 1B LLM by substituting the output layer into a classification layer. we construct our dataset with over 3 million safety preference data and train our reward model until it achieved complete convergence. After extensive experimentation, we find that the offline safety score (ADPO) in natural flow has improved significantly (0.9020 to 0.9670), primarily attributed to the improvement in the EHD data (0.9430 to 0.9705), while the precise of the model didn't show a decent decline (0.5185 to 0.5180). Furthermore, after a month of online iterations, the safety score has further improved to 0.9855, The experiment results suggest that our

approach will only affect data directly related to harmful facts, without significantly compromising the helpfulness of the LLMs.

## 5 RELATED WORKS

**LLM Safety alignment.** Ensuring the safety and ethical alignment of LLMs is a critical area of research, with Supervised Fine-tuning (SFT) and Reinforcement Learning from Human Feedback (RLHF) emerging as key methodologies. SFT involves refining a pre-trained model on a curated dataset that emphasizes desirable behaviors, which can significantly improve the model's adherence to ethical guidelines and reduce the likelihood of generating harmful or biased content (Ouyang et al., 2022; Ziegler et al., 2019; Brown et al., 2020). For instance, Ouyang et al. (2022) demonstrated that by training LLMs on a dataset of instructions and their corresponding desired outputs, the models could better follow user commands and produce more aligned responses. Similarly, Ziegler et al. (2019) showed that fine-tuning on a smaller, high-quality dataset can enhance the model's ability to generate text that aligns with specific ethical standards. Complementing SFT, RLHF leverages human evaluators to provide direct feedback on the model's outputs, guiding the learning process towards more ethically aligned and contextually appropriate responses (Stiennon et al., 2020; Sastry et al., 2020; Christiano et al., 2017; Baird et al., 2022; Schick & Schütze, 2023). Stiennon et al. (2020) introduced a method for training summarize models using human feedback, showing that this approach can lead to summaries that are more accurate and coherent. Sastry et al. (2020) further extended this work to align LLMs with human values, emphasizing the importance of continuous human oversight in the training process. Additionally, Christiano et al. (2017) explored the use of human preferences to shape the behavior of reinforcement learning agents, which has been adapted for LLMs to ensure that they learn to prioritize human-aligned outcomes. Recent studies have also focused on integrating SFT and RLHF to create more robust and aligned LLMs. Baird et al. (2022) presented a framework for scaling RLHF to complex tasks, demonstrating that with proper reward modeling, it is possible to achieve high-quality performance even in scenarios with sparse rewards. Schick & Schütze (2023) further refined the reward functions used in RLHF, enhancing the model's ability to capture nuanced human judgments. Other works, such as Wang et al. (2022); Zhang et al. (2022); Liu et al. (2022), have explored various techniques to improve the effectiveness of RLHF, including the use of diverse human feedback, advanced reward modeling, and multi-objective optimization. These studies collectively highlight the ongoing efforts to develop safer and more responsible AI systems, underscoring the importance of an iterative and interactive approach to alignment, where continuous human feedback plays a crucial role in mitigating risks and fostering trust in LLMs.

**Balance between helpfulness and harmlessness.** Several works focus at finding the balance between functionality and safety. Ji et al. (2024a) incline to modify the response during the reasoning process and propose an external residual model aligner. Dai et al. (2023) focused on the alignment process and proposed a Safe-RLHF training framework. This line of inquiry emphasizes the importance of building models that can robustly respond to unforeseen circumstances without compromising their operational efficacy. Several works also tried to find a optimized method to do safety alignment through psychological ways Heston (2023); Wu et al. (2024) and red teaming Ge et al. (2023); Perez et al. (2022); Ganguli et al. (2022).

## 6 CONCLUSION

In conclusion, this paper investigates the underlying causes of risks associated with LLMs and proposes a novel safety system designed to find a balance between the helpfulness and harmlessness of these models. Our approach encompasses three critical dimensions: data management, training architecture, and external protective measures. Experimental results demonstrate that our method significantly outperforms existing solutions. Future work will extend our findings from the textual domain to the field of Multi-modal Large Language Models (MLLM), such as LLava Liu et al. (2024), Visual Question Answering(VQA) Antol et al. (2015); Goyal et al. (2017) model, and audio LLM Lyu et al. (2023).

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
