# OpenReview forum: "Truly Safe & Truly Helpful: Achieving Harmonious Balance for Large Language Model"
_ICLR.cc/2025/Conference — Submitted to ICLR 2025_

### Official Review · Reviewer_buFa · 2024-10-30

**Soundness:** 2
**Presentation:** 1
**Contribution:** 2
**Rating:** 3
**Confidence:** 4

**Summary:**

The paper proposed a few different things:
1. Safety training data preparation
2. message-wise alignment training
3. a harmful token filtering mechanism applied during the inference phase.

Starting with the claim that safety concerns in LLMs can be due to (1) inadequate safety alignment or (2) insufficient safety knowledge.
The paper argues that safety alignment training is to teach the model to better interpret the internal reason of a risk, rather than learning new safety knowledge. As simply increasing the quantity of safety data (with high quality and diversity) does not consistently lead to significant improvement in models’ safety.
Split into
- EHD (explicit harmful data): factual risk data; influenced by internal knowledge
- IHD (implicit harmful data): intentional risk data, wo explicit risky content; valuable data for safety alignment
- MHD (mixed): explicit risk content and malicious intent; impacted by both knowledge and alignment
Safety scores on different datasets saturated at different levels.

Adaptive message-wise PPO training applies a masking function where only samples in $Y_w$ (chosen set) with higher than baseline reward or in $Y_l$ (rejected set) with lower than baseline reward. This masking can be applied to PPO, DPO, etc.

Harmful token filtering at inference time can be done by:
1. search tokens at inference time based on a safety reward model's score
2. a RAG framework with a dataset of harmful entities and try to avoid generating them at inference time.

Experiments were conducted on each of these three things and conclusions feels a bit hand-wavy:
- Facts and intent reinforce mutually in safety alignment.
- More data does not means no safe.
- Truly safety requires truly understanding.
- Adaptive methods brings great general performances.

**Strengths:**

The idea of splitting safety alignment and safety knowledge is interesting.
It is also an interesting idea to split data according to different safety features and then evaluate or train the model on each or experiment with different mixtures.
However, more work should be done here.

**Weaknesses:**

Overall, this paper feels like a collection of random ideas. The common theme might be about this claim around safety alignment behavior vs safety knowledge, but the experiment design is not clear enough. And other parts on adaptive training or inference time filtering feels a bit off topic.

Other than the data categorization idea, I don't quite understand how other two parts are connected. I'm also not convinced by the results of applying adaptive masking is necessary; e.g. is it possible the lifting is just due to better quality data points?

The harmful token filtering at inference time part should be cut.

**Questions:**

This paper needs a major rewrite.

---

> ### Author Response · Authors · 2024-11-18
> **Rebuttal for Weaknesses**
>
> Dear Reviewer,
>
> Thank you for your thoughtful review and the opportunity to clarify our work. We appreciate your feedback and would like to address the concerns you've highlighted.
>
> Our primary aim in this paper is to present a comprehensive safety framework for large language models that addresses both malicious input issues and risk entity-related concerns. In practical applications, language models often encounter unsafe scenarios. We categorize these into two types: harmful content arising from malicious intent and factual risks arising from prohibited sensitive vocabulary.
>
> Regarding your comment about the coherence of the different components, we would like to emphasize that the three proposed methods are designed to address distinct facets of these safety issues. No single approach, such as alignment alone, can effectively mitigate all potential risks. In fact, merely increasing safety-aligned data can lead to an unintended consequence of models becoming overly cautious and refusing legitimate queries, thus degrading their utility.
>
> Data Categorization: We propose categorizing data based on intent and fact, allowing for more effective safety alignment with fewer data. This method ensures that models internalize safer value systems by focusing on specific types of risks separately.
>
> Adaptive Masking: Traditional reinforcement learning methods often result in a loss of information, failing to achieve true alignment. Our masking strategy mitigates this issue, allowing the model to align more securely without sacrificing informative content.
>
> Harmful Token Filtering: Post-alignment, we identified gaps in the model's recognition of certain risk entities, such as politically sensitive terms banned by the Chinese government. The sheer volume of such terms—potentially in the millions—makes alignment insufficient. Thus, we introduced an external harmful token filtering mechanism to address these risks effectively, providing an additional layer of safety.
>
> We understand the need for detailed experiments to substantiate the necessity and effectiveness of adaptive masking beyond just high-quality data. We will further clarify these aspects in the manuscript to demonstrate the integration and significance of each method within our safety framework.
>
> We hope this explanation addresses your concerns about the interconnectedness and relevance of the different sections of our work. We are grateful for your feedback, which enables us to improve the clarity and impact of our research.

---

### Official Review · Reviewer_8yWK · 2024-11-02

**Soundness:** 2
**Presentation:** 2
**Contribution:** 2
**Rating:** 5
**Confidence:** 4

**Summary:**

This work first proposes to distinguish safety knowledge from safety values. Then, it proposes a pipeline including data preparation, training, and external risk filtering. Using this pipeline, this work claims that a better trade-off between safety and harmfulness can be obtained.

**Strengths:**

- The notions of safety knowledge and safety value are novel.
- This work proposes a comprehensive pipeline which covers pre-processing, alignment, and post-processing. This could act as a mature solution for the industry.
- Extensive experiments are conducted to verify the three steps.

**Weaknesses:**

- As they are mostly defined in natural language, the concepts of safety knowledge and safety value are not very clear. I also checked the examples in the appendix; but did not understand the difference between the first (EHD; Political) and the second (IHD; Political). *To better distinguish these notions, it would be helpful to provide concrete examples in the main manuscript.*
- It is unclear how the proposed method/pipeline serves to make a better trade-off between safety and helpfulness. For example, in Figure 2 and Figure 3, it is unclear how different safety data distributions affect the helpfulness.
- There are several unconvincing claims:
1, line 238, ‘This highlights ... restrict their safety’
2, line 356, ‘RAG ... assesss contextual harm’
3, line 391, ‘This indicates ... the knowledge it possesses’

**Questions:**

- How is Equation (13) connected to other contents in the manuscript?
- How is RAG used in section 3.3? From Equation (14), it is unclear how RAG works.
- Is there anything wrong with the “More data does not means no safe” in line 394. I do not understand how it is connected to the following analysis.
- How are ‘anti-risk-intent’ and ‘anti-risk-fact’ defined in line 382?

---

> ### Author Response · Authors · 2024-11-18
> **Rebuttal on Weakness**
>
> Dear Reviewer,
>
> Thank you for your thoughtful feedback on our manuscript. We appreciate your insights and would like to address the points raised regarding the clarity of concepts and the demonstration of our method's efficacy.
>
> Regarding the first comment, we have clarified in the manuscript that Explicit Harmful Data (EHD) refers to factual risk issues, such as explicitly prohibited political terms or other risk entities. Implicit Harmful Data (IHD), on the other hand, involves data with harmful intent without explicit banned entities, such as red team attacks, sarcasm, and insinuations. We understand the need for concrete examples to distinguish these concepts more clearly. However, many EHD examples involve explicitly banned content, and presenting them could raise ethical or political issues. To address this, we will consider adding more illustrative, general examples in the main manuscript that can be provided without ethical concerns.
>
> For the second comment, our manuscript's pipeline is designed to address the limitations of relying solely on alignment methods for safety issues. Simply incorporating more safety data may not enhance model safety; instead, it may lead to unnecessary refusals for legitimate queries. Our approach seeks to optimize both the model's value alignment and its knowledge base regarding risks, ensuring enhanced safety without compromising general utility. This is why we have designed three distinct methods:
>
> Data Categorization: By categorizing data through intent and fact perspectives, we can achieve better safety alignment with less data, forming a more secure value system within the model.
>
> Adaptive Masking: Traditional reinforcement learning approaches can cause significant information loss, hindering effective alignment. Our masking strategy mitigates this, promoting better value alignment.
>
> Harmful Token Filtering: Post-alignment, we identified a gap in the model's awareness of risk entities, especially those with sensitive political connotations banned by entities like the Chinese government. These terms often exceed the model's knowledge base and cannot be fully managed by alignment alone due to their vast number. This necessitated our third method, an external harmful token filtering mechanism.
>
> Figures 2 and 3 are intended to illustrate the balance between safety and helpfulness. We will provide additional commentary and examples to clarify how different safety data distributions impact both metrics.
>
> We trust that this response clarifies the connections between safety knowledge, value, and the necessity of our proposed methods. We appreciate the opportunity to refine our presentation and ensure a comprehensive understanding of our contributions.

---

### Official Review · Reviewer_HfZB · 2024-11-04

**Soundness:** 3
**Presentation:** 2
**Contribution:** 3
**Rating:** 6
**Confidence:** 2

**Summary:**

The paper studies alignment of LLMs, through the lens of different types of harmful data. An optimal data distribution is found empirically, in terms of the number of harmful data points from each type. This is combined with adaptive preference optimization and token-filtering during inference to improve upon current methods of alignment. The authors conduct a thorough evaluation of their proposed methods relative to existing ones and find an improvement in safety while generally maintaining similar levels of helpfulness. The empirical results presented in the paper on the dynamics of LLM alignment during training and the effects of different types of harmful data are insightful.

**Strengths:**

- Topic of interst: Alignment that incorporates properly both safety and helpfulness is important.
- Interesting experimental results on safety vs helpfulness due to finetuning - for example, saturation of safety score and drop of helpfulness score as more safety training data is introduced.
- Categorization of types of harmful data, their impact on the model, and a proposal for adjusting their distribtution (ratio to general training data) to achieve better performance.
- Strategy of highlighting segments based on a reward during training is an interesting approach. The experiments show a significant increase in safety with the adaptive approach compared to existing approaches, while maintaining similar levels of helpfulness.
- Risk token filtering looks like an interesting approach to safety at inference time - given that adversarial attacks can be very short, and specific words can trigger unsafe behavior, this seems like a good approach to study.

**Weaknesses:**

- Safety score on explicit harmful data remains low - while the adaptive approaches are higher than the baselines, they do not significantly improve. Just as a reference, it would be interesting to see the performance of more well-known models of similar size on this metric, e.g. llama-3-8B.
- Partial results on token filtering - It seems the results on token-filtering are only reported in average safety and average "precision", without reference to the full results (not in the main text or appendices). It is hard to understand exactly what the results are based on these reports.
- (Minor) - typos - there are quite a few in the paper (e.g. figure 2 - "Number of safety training data" -> "Number of examples(?) in safety training data", line 394 - "More data does not means no safe" -> "More data does not mean not safe").
- (Minor) - in line 387, there is a reference to a figure 5 (which does not exist), I assume it is for figure 3.
- Related works - There are additional works on balancing helpfulness and harmlessness that can be mentioned, e.g. [1,2,3].
- Overall I am impressed with the results of the paper, which provide very interesting insights, but feel that the technical issues mentioned above accumulate, so they need to be addressed.


[1] - https://arxiv.org/abs/2309.07875

[2] - https://arxiv.org/abs/2308.01263

[3] - https://arxiv.org/abs/2401.16332

**Questions:**

- For reference, how would more well-known models of similar sizes perform on the safety metrics?
- Is there a way to improve the safety score on explicit harmful data?

---

> ### Author Response · Authors · 2024-11-18
> **Rebuttal for experiment results**
>
> Dear Reviewer,
>
> Thank you for your insightful feedback on our manuscript. We appreciate your comments and would like to address the two main concerns you've raised.
>
> Firstly, regarding the safety score on explicit harmful data (EHD), we recognize that while our adaptive approaches show improvements over the baselines, they do not fully resolve safety issues on EHD. This aligns with our key motivation: alignment alone cannot completely address the challenges posed by EHD due to the vast and potentially infinite number of risk entities. This limitation underscores the necessity of our proposed harmful token filtering, an external method designed to tackle the problem of risk entities more effectively.
>
> Secondly, with respect to your suggestion about using more well-known models such as llama-3-8B for benchmarking, the primary reason we did not include these models is rooted in our focus on a Chinese dataset. A significant portion of the risk entities in our study pertains to sensitive political and legal issues specific to China, which far exceed the knowledge base of English pre-trained models like llama-3-8B. Consequently, we selected Qwen as our foundational model for experimentation due to its more robust handling of these context-specific challenges.
>
> Regarding the comment about the partial results on token filtering, we apologize for any confusion. Our intention was to focus on average safety and precision metrics to provide a succinct overview. However, we understand the need for comprehensive data and will ensure that detailed results, including full metrics, are clearly presented in the revised manuscript and appendices to facilitate a better understanding of our findings.
>
> We trust this clarifies our approach and the choices made in our experimental design, and we appreciate your consideration of our responses. Thank you once again for your constructive feedback, which helps us enhance the quality and clarity of our work.

---

> > ### Comment · Reviewer_HfZB · 2024-11-26
> >
> > I would like to thank the authors for their clarifications. I believe the score is suitable, as the paper contains interesting results, but the clarity is somewhat lacking due to the writing issues mentioned above and by the other reviewers.

---

### Official Review · Reviewer_MH2Q · 2024-11-04

**Soundness:** 3
**Presentation:** 2
**Contribution:** 2
**Rating:** 3
**Confidence:** 4

**Summary:**

This paper proposes three approches to improve the tradeoff between alignment and helpfullness in LLMs: (1) identify data in different categories, (2) mask the gradients of the less significant segments, and (3) decode with additional filtering process. Experiments show that the proposed approches improves the alignment and the helpfulness compared to baselines.

**Strengths:**

- The topic of LLM alignment is interesting and important.
- The structure of the paper is clear.

**Weaknesses:**

- The writing of the paper is not clear enough, making it hard to follow. For example, the paper fails to mention the references to the supplementary materials.
- The paper is missing lots of details, making it hard to understand the proposed approches and justify the advantages of the methods. For example, the detailed settig of figure 2 is missing. What are dataset a and b? How do we define the safety score? What is the real-world data? etc. In algorithm 1, how are the forbidden words defined? How did the authors collect the 260k data? Section 4.3 is missing lots of details as well.
- The contribution of the paper is limited. The paper seems like 3 different tricks combined together without correlations for mutual improvements. The 3 tricks, including seperating data categories, masking gradients, and adding loss during decoding are all studied in previous work.

**Questions:**

Please see the weaknesses above.

---

> ### Author Response · Authors · 2024-11-18
> **Rebuttal for the motivation and contribution of this paper**
>
> Dear Reviewer,
>
> We appreciate your thorough review and valuable feedback on our manuscript. We would like to address your concerns regarding the perceived lack of correlation among the three methods we proposed (separating data categories, masking gradients, and adding loss during decoding).
>
> The core motivation of our work stems from challenges encountered in practical applications of large language models (LLMs), particularly in ensuring their safety alignment. Through our observations, we identified that model safety alignment is influenced by the knowledge base of the model, its understanding capabilities, and its alignment mechanisms. Traditional alignment methods typically augment the models with more and higher-quality safety-aligned data without delving into the underlying mechanisms. This approach, however, often fails to achieve effective safety alignment and inadvertently hampers the model's general capabilities due to the increased volume of safety data.
>
> To explore these internal relationships more profoundly, we divided the issues into two categories at the data level and conducted separate alignment experiments and analyses. This explains our rationale for employing the data classification method. While solely classifying data can improve safety, it doesn't yield optimal results in practical applications. Firstly, it struggles to balance safety and the model's false rejection rate in alignment tasks. Secondly, given the vast array of potential risk facts, alignment alone cannot effectively preempt all risks. Thus, we introduced the adaptive message-wise alignment approach and the harmful token filtering mechanism to achieve better safety outcomes while reducing the model's false rejection rate.
>
> Regarding your observation that these three methods have been previously mentioned in other works, we would like to clarify that this is a misinterpretation. Our manuscript is the first to propose the classification of all training and benchmark data into intents and facts, which is a novel approach. Furthermore, the adaptive message-wise alignment method significantly differs from traditional token-level DPO, as evidenced by our experiments demonstrating its advantages over token-DPO. Lastly, while similar methods to the harmful token filtering mechanism have been discussed in previous literature, our manuscript is the first to apply this approach specifically for filtering harmful tokens. Additionally, we addressed the iterative application challenges in practical scenarios by incorporating RAG (retrieval-augmented generation) and leveraging an offline knowledge base.
>
> We hope our explanation clarifies the novelty and the systematic correlations among the methods we present, and we appreciate your consideration of our detailed responses.

---

> > ### Comment · Reviewer_MH2Q · 2024-12-03
> >
> > I want to thank the authors for their response. However, the authors only answered one of my questions, and I didn't see any updates for other questions. The response above only addresses part of my concern, I still don't see the correlation between different tricks. I think the paper would benefit from experiments to show the mutual improvements and connections between the tricks, otherwise, it seems like putting 3 tricks together, as also pointed out by the other reviewers. I'll keep my score for now.

---

### Meta-Review · Area_Chair_YA2k · 2024-12-20

**Metareview:**

The submission presents several techniques to improve the trade off between alignment and helpfulness of LLMs.

+ The topic is very relevant to the machine learning community.
+ Some of the ideas presented were interesting to the reviewers.

- The paper is not clearly written.
- The various techniques presented do not have a common theme, but are instead of collection of unconnected ideas.

**Additional Comments On Reviewer Discussion:**

While one of the reviewers (with low confidence) recommends acceptance, the other reviewers recommend rejection. The author rebuttal was carefully considered. However, the reviewers agree that it does not address the issues raised in the initial reviews.

---

### Decision · Program_Chairs · 2025-01-22

Reject